# Tandem Mass Tag-Based Quantitative Proteomic Analysis of ISG15 Knockout PK15 Cells in Pseudorabies Virus Infection

**DOI:** 10.3390/genes12101557

**Published:** 2021-09-30

**Authors:** Wenfeng He, Chen Li, Liangliang Dong, Guoqing Yang, Huimin Liu

**Affiliations:** College of Life Science, Henan Agricultural University, Zhengzhou 450002, China; hewenfeng0825@163.com (W.H.); lichen960724@163.com (C.L.); dll1103@163.com (L.D.); gqyang@yeah.net (G.Y.)

**Keywords:** ISG15, PRV, TMT, proteomic analysis, DEPs

## Abstract

Pseudorabies virus (PRV) is recognized as one of the most important pathogens of swine and poses a serious threat to the swine industry worldwide. Available commercial vaccines fail to protect against the emergence of new PRV strains. Therefore, the new protein targets against PRV highlight the urgent need for uncovering the molecular determinants of host cellular proteins following PRV infection. Interferon-stimulated gene 15 (ISG15) demonstrates an outstanding antiviral response. However, the molecular mechanism of ISG15 that affects PRV replication is incompletely known. Here, we performed a tandem mass tag (TMT)-based approach to quantitatively identify protein expression changes in PRV-infected ISG15 knockout PK15 (ISG15^−/−^-PK15) cells. In total, 4958 proteins were identified by using TMT coupled with LC-MS/MS in this study. In the PRV- and mock-infected groups, 241 differentially expressed proteins (DEPs) were identified, 162 upregulated and 79 downregulated proteins at 24 h post-infection (hpi), among which AFP, Vtn, Hsp40, Herc5, and Mccc1 may play important roles in PRV propagation. To ensure the validity and reliability of the proteomics data, the randomly selected DEPs were verified by RT-qPCR and Western blot analysis, and the results were consistent with the TMT results. Bioinformatics analyses further demonstrated that the DEPs are mainly involved in various biological processes and signaling pathways, such as signal transduction, the digestive system, and the PI3K-AKT pathway. These findings may provide new insight into molecular mechanisms for PRV infection, which is helpful for identifying potential protein targets for antiviral agents.

## 1. Introduction

Pseudorabies (PR), an acute, febrile infectious disease caused by pseudorabies virus (PRV), which is a member of the subfamily Alphaherpesvirinae in the family Herpesviridae [1]. PRV was discovered at the beginning of the 20th century and is prevalent in at least 44 countries [2]. PRV can cause a high mortality rate in newborn pigs, neurological and respiratory system disorders in growing pigs, and reproductive failure in pregnant sows [3]. Pigs are the only natural hosts for PRV and are the main source of its transmission, although PRV can infect a broad range of hosts, including livestock and wildlife [2,4,5]. More recently, it has been reported that PRV can infect humans with weakness, fever, sweating, dysphagia, and neurological dysfunction as typical symptoms [6,7,8,9]. Since 2011, highly pathogenic PRV variants that emerged in several pig farms in China have caused substantial economic losses to the pig industry worldwide [10]. Thus, the threat of PRV should not be ignored, and new targets are urgently needed to control PRV spread are urgently needed.

The interactions between viruses and host cells are extremely complex because they involve many proteins and host cellular pathophysiological processes. Viruses can subvert host protein expression to facilitate their infection, and host cells have evolved numerous strategies to inhibit viral replication and establish an antiviral state. Despite years of intensive research, the mechanisms of PRV pathogenesis and immunomodulation remain largely unknown [11,12,13,14]. Accumulating evidence emphasizes that proteomic techniques have been increasingly widely used to screen the differentially expressed proteins associated with viral replication, an approach helpful for further revealing the molecular mechanism of virus-host cell interactions involved in viral pathogenesis [14,15,16].

To date, it appears that very few studies have been performed to analyze the interplay between PRV and host cells using proteomics analysis. Here, a quantitative proteomic approach of tandem mass tags (TMT) labeling coupled with LC-MS/MS was applied to analyze the interplay between PRV and host cells, an approach that could provide a better understanding of PRV pathogenesis and the potential antiviral targets against PRV. Interferon-stimulated gene 15 (ISG15), one of the most strongly induced ISGs during viral infection, was found in our previous study to play an antiviral role in PRV replication [17]. However, the role and mechanism of ISG15 in PRV infection has not been explored. ISG15 can modify a diverse array of biological processes and targets by a posttranslational modification known as ISGylation [18]. Therefore, an ISG15 knockout PK15 cell line (ISG15^−/−^-PK15) was generated and used as an ideal cell model to identify the proteins associated with ISG15 during PRV infection. Our findings provide valuable information for understanding the ISG15 knockout-induced proteome changes following PRV infection, and these findings are expected to shed some light on potential antiviral targets and signaling pathways involved with PRV pathogenesis. 

## 2. Materials and Methods

### 2.1. Cells and Virus 

Porcine kidney epithelial (PK15) cells were cultured in Dulbecco’s Modified Eagle Medium (DMEM) supplemented with 10% fetal bovine serum and 1% antibiotics and were maintained at 37 °C in 5% CO_2_. The PRV QXX strain used in this study was isolated and stored in our laboratory [18]. When PK15 cells were 80% confluence, they were washed three times with sterile phosphate-buffered saline (PBS, pH 7.2). Then, 100 μL of the PRV strain was inoculated at a multiplicity of infection (MOI) of 2. After 1 h of adsorption, infected cells were maintained in DMEM supplemented with 2% FBS. Uninfected cells served as the mock-infected group. The PRV- and mock-infected cells were harvested at 24 h post-infection (hpi). Three independent technical replicates of each group were established. Viral propagation was confirmed by assessment of the cytopathic effect (CPE) under a light microscope and constructing a one-step growth curve of PRV at 6, 12, 24, and 36 hpi. 

### 2.2. Establishment of the ISG15^−/−^-PK15 Cell Line

The porcine ISG15 sequence was searched in Genebank and ISG15 (gene identification number NM_001128469) was analyzed. Based on the sequence of the single exon, the two specific guide RNA sequences (sgRNAs) were designed using the CRISPR web tool (http://crispor.tefor.net/crispor.py, accessed on 15 March 2020). The sgRNA sequences were cloned into the digested pX459 vector with FastDigest BbsI (Thermo Fisher Scientific, Inc., Waltham, NY, USA), and the vector was then transfected into HEK293 cells for 48 h. The pX459-ISG15 was transformed into PK15 cells and clones were selected by screening with puromycin (2 μg/mL) for 2 days and were then seeded at 1 cell per well in 96-well plates. Positive clones were screened and sequenced (Shanghai Shenggong Technology Co, Ltd., Shanghai, China). Total protein of ISG15 knockout PK15 cells was extracted and the expression of ISG15 was detected by Western blot analysis. Subsequently, an MTT assay was used to evaluate the effect of ISG15 knockout on PK15 cell proliferation.

### 2.3. Immunofluorescence Assay 

ISG15^−/−^-PK15 cells at different time points after infection with PRV (MOI = 2) and mock-infected cells at 24 h were washed three times with sterile PBS. Then, the cells were fixed with 4% paraformaldehyde for 20 min and permeabilized with 0.2% Triton X-100 (Solarbio Life Science, Beijing, China) for 15 min at room temperature (RT). After washing with PBS three times, the cells were blocked with PBS containing 5% bovine serum albumin for 30 min at RT. After three washes with PBS, the cells were incubated with anti-PRV-gE mouse monoclonal primary antibody (prepared and stored in our laboratory) at RT for 2 h. After being washed with PBS, the cells were incubated with an Alexa Fluor 488 conjugated goat anti-mouse secondary antibody (Proteintech, Wuhan, China). Cell nuclei were stained with 0.01% 4′, 6-diamidino-2-phenylindole (DAPI) (Solarbio Life Science, Beijing, China). Fluorescence images were acquired using an inverted fluorescence microscope (Olympus, Tokyo, Japan).

### 2.4. Protein Extraction, Digestion, and TMT Labeling

PRV- and mock-infected ISG15^−/−^-PK15 cell samples at 24 hpi were collected and resuspended in lysis buffer (4% SDS, 1 mM DTT, 150 mM Tris-HCl (pH 8.0), protease inhibitor) after centrifugation. Then, the samples were sonicated on ice for 3 min, and cellular debris was removed by centrifugation (16,000 *g*/min, 10 min, 4 °C). The protein concentration in the extracts was determined with a BCA protein quantitation kit (Beyotime). Protein digestion was performed according to a previously described procedure [16]. Proteins in solution were reduced with 5 mM DTT at 56 °C for 30 min and alkylated with 11 mM IAA for 15 min at RT in the darkness. The samples were then diluted to a urea concentration of less than 2 M by the addition of 100 mM TEAB. Then, trypsin was added at a 1:50 trypsin-to-protein mass ratio for overnight digestion and at a 1:100 ratio for a second 4 h digestion. After trypsin digestion, the peptides were desalted on a Strata X C18 SPE column (Phenomenex, Torrance, CA, USA) and vacuum dried. The peptide mixture was recombined in 0.5 M TEAB and processed according to the TMT kit protocol, and sequentially incubated for 2 h at RT, pooled, desalted, and dried by vacuum centrifugation. For labeling, the dried peptide mixture was reconstituted and acidified with 2 mL of buffer (10 mM KH2PO4 in 25% ACN, pH 2.7) and loaded onto a Poly SULFOETHYL 4.6 × 100 mm column (5 mm, 200 Å, PolyLC Inc., MD, Columbia, USA). Peptides were eluted at a flow rate of 1 mL/min with a gradient of 0–10% for 2 min, 10–20% to 25 min, 20–45% to 32 min, 45–100% to 42 min, and at 100% from 47–52 min and 52–60 min. The eluted fractions were collected every 1 min and were then combined into 10 pooled fractions and desalted on C18 cartridges (EmporeTM SPE Cartridges C18, Sigma, Roedermark, Germany). Each fraction was concentrated by vacuum centrifugation and reconstituted in 40 mL of 0.1% trifluoroacetic acid. All samples were stored at −80 °C for LC-MS/MS analysis.

### 2.5. LC-MS/MS Analysis 

The LC–MS/MS analysis was performed as described previously [19]. Approximately 10 μL of each fraction was injected for nanoLC-MS/MS analysis. The peptide mixture (2 µg) was loaded onto a Thermo Scientific EASY trap column and separated with a linear gradient of solvent B (100% for 40 min, 28–90% for 2 min, and 90% for 18 min). The mass spectrometer was operated in a positive ion mode and MS spectra were acquired. The data were acquired using a data-dependent top 10 method, and ions were fragmented through high-energy collisional dissociation. Determination of the target value was based on predictive automatic gain control (pAGC). The normalized collision energy was 30 eV and the underfill ratio, which specifies the minimum percentage of the target value likely to be reached at the maximum fill time, was defined as 0.1%. 

### 2.6. Bioinformatic Analysis 

The obtained MS/MS data were searched against the UniProt Sus scrofa database for peptide identification and quantification using the Proteome Discoverer Software 2.1 (Thermo Fisher Scientific, Waltham, NY, USA). The quantitated proteins were those that were found in three biological replicates, and at least two peptides had to have a confidence value higher than 95% for protein quantification. To guarantee the accuracy of the MS data analysis, the cutoff peptide and protein confidences values were set to >95% and >1.2, respectively, coupled with a false discovery rate (FDR) of ≤1% for peptide and protein identification. 

Differentially expressed proteins (DEPs) were screened by the criteria of a 1.2-fold increase or 0.83-fold decrease, and a *p*-value < 0.05. Gene ontology (GO) and Kyoto Encyclopedia of Genes and Genomes (KEGG) enrichment analyses of these DEPs and protein–protein interaction (PPI) mapping (based on STRING, http://string-db.org/, accessed on 2 November 2020) were performed with the OmicsBean software suite (http://www.omicsbean.cn/, accessed on 2 November 2020).

### 2.7. Validation of Targeted Proteins by RT-qPCR and Western Blot Analysis

To verify the reliability of the DEPs identified by the TMT labeling approach, the mRNA and protein levels of the randomly selected proteins were determined by RT-qPCR and Western blot analysis, respectively. The sequence information of the primers used for RT-qPCR is shown in Table 1. All data are presented as the means and standard deviations (SDs), and the statistical significance of differences is denoted as *** *p* < 0.001, ** *p* < 0.01, and * *p* < 0.05. All experiments were reproducible and performed in triplicate, and the data were analyzed using GraphPad Prism 6 software (GraphPad Software, Inc., San Diego, CA, USA).

## 3. Results

### 3.1. Establishment of the Porcine ISG15^−/−^-PK15 Cell Line

To explore the role and mechanism of ISG15 in PRV replication, we constructed an ISG15 knockout PK15 cell line using CRISPR/Cas9 gene-editing technology. The recombinant plasmid of PX459-ISG15 was successfully constructed and was then transfected into the PK15 cells. The monoclonal cell line was selected using puromycin (2 μg/mL). Gene sequencing showed that the ISG15 knockout PK15 cell line was successfully constructed, and the knockout efficiency was evaluated by Western blot analysis. Compared with that in the wild type PK15 cell (WT) group, the expression level of ISG15 in the ISG15-knockout sgRNA-1 and sgRNA-2 groups was significantly decreased, suggesting that ISG15 was completely silent in PK15 cells (Figure 1A). Moreover, there was no significant difference in the ISG15 expression level between the ISG15 knockout sgRNA-1 group and the sgRNA-2 group. Subsequently, the gene sequence alignment revealed that the sgRNA-1 and sgRNA-2 cells were heterozygous for the ISG15 knockout alleles, and these cells were designated the ISG15^−/−^-PK15 cell lines. In addition, the effect of ISG15 deficiency on the proliferation of PK15 cells was evaluated by an MTT assay. The results revealed that there was no significant difference in cell proliferation between the WT and ISG15^−/−^-PK15 groups at different time points (Figure 1B). Taken together, these results indicated that the ISG15^−/−^-PK15 cell line, which could provide an ideal cell model to screen the new targets associated with ISG15 during PRV replication, was successfully constructed.

### 3.2. Determination of the Optimal Time for Proteomic Analysis Following PRV Infection in ISG15^−/−^-PK15 Cells

To determine the kinetics of PRV propagation and the optimal time point for proteomic analysis, ISG15^−/−^-PK15 cells were infected with PRV (MOI = 2), and the CPE was monitored by microscopy at 6, 12, 24, and 36 hpi. In addition, viral titers and viral protein expression were detected at the above time points. As shown in Figure 1C, no obvious CPE was observed at 6 hpi in PRV-infected ISG15^−/−^-PK15 cells; however, the CPE gradually became increasingly apparent as the infection progressed. An obvious CPE was observed at 12 hpi and became more evident at 24 and 36 hpi; the manifestations included cell rounding, cell enlargement, cell detachment, and granular degeneration of the cytoplasm, and the majority of the cells developed a severely damaged morphology and floated in the medium beginning at 36 hpi. The one-step growth curve of PRV revealed that the viral titer peaked at 24 hpi and then gradually declined (Figure 1C). The expression of PRV proteins was further monitored by IFA, and the results showed that almost all cells became infected at 24 hpi (Figure 1D). The time point when viral replication remains high is often regarded as the optimal time for proteomic analysis [16]. Therefore, in consideration of cell integrity and the high infection rate of the cells, 24 hpi was selected as the optimal time point for further proteomic analysis.

### 3.3. PRV-Induced Changes in Protein Expression in ISG15^−/−^-PK15 Cells

To comprehensively analyze the changes in host cells during PRV infection, TMT labeling coupled with LC–MS/MS analysis was performed in ISG15^−/−^-PK15 cells (Figure 2A). A total of 4958 peptides were identified by spectral analysis in both PRV- and mock-infected ISG15^−/−^-PK15 cells at 24 hpi. Based on the two criteria of a fold change >1.2-fold or <0.83-fold and a *p*-value < 0.05, 241 DEPs were identified, among which 162 were markedly upregulated and 79 were significantly downregulated in PRV-infected ISG15^−/−^-PK15 cells compared with the mock-infected cells (Appendix A). In the volcano plot, upregulated DEPs are represented by red dots, and downregulated DEPs are represented by blue dots (Figure 2B). Additionally, the DEPs in each group were analyzed and shown in a hierarchical clustering heat map (Figure 2C).

### 3.4. Validation of DEPs by RT-qPCR and Western Blot Analysis

To ensure the reliability of the obtained proteomic data, seven proteins (AFP, Hsp40, Herc5, Mccc1, Vtn, Strap, Fn1 and IL18) were randomly selected for verification through RT-qPCR and Western blot analysis; these proteins represented the sets of upregulated and downregulated proteins. In addition, the PRV-gE protein was chosen to verify PRV replication. As shown in Figure 3, RT-qPCR and Western blot analysis of these proteins in PRV- and mock-infected ISG15^−/−^-PK15 cells showed results consistent with those of TMT quantitative proteomic analysis (Figure 3). These results indicate the high credibility of the DEPs identified using the TMT labeling approach.

### 3.5. GO Enrichment Analysis of DEPs

The DEPs were assigned to the three main categories of Gene Ontology (GO) categories: biological process (BP), cellular component (CC), and molecular function (MF) after protein function analysis. We used the GO analysis to reveal the functional implications of the identified DEPs in the PRV-infected groups. These proteins were markedly enriched in terms of single-organism process, metabolic process, immune system process, multicellular organismal process, developmental process, etc., in the BP category. In the CC category, these proteins were significantly enriched in the terms cell part, organelle, extracellular region, supramolecular fiber, and cell junction, and so on. In the MF category, these proteins were observably enriched in the terms binding, catalytic activity, molecular function regulator, structural molecule activity, transporter activity, and antioxidant activity (Figure 4A).

All upregulated and downregulated proteins were further subjected to GO enrichment analysis, and each protein was found to belong to at least one term. Among the upregulated proteins, the three major functional classes (>40% for each class) of the identified proteins in the BP category were single-organism process (101 proteins), metabolic process (91 proteins), and multicellular organismal process (54 proteins). The major classes in the CC category were cell part (105 proteins), organelle (101 proteins) and organelle part (60 proteins) (>40%). The major functional class in the MF category was binding (149 proteins) (>70%) (Figure 4B). Among the downregulated proteins, the main terms in the BP category were single-organism process (53 proteins), metabolic process (54 proteins), and multicellular organismal process (32 proteins). The main term in the MF category was binding (60 proteins). The main terms in the CC category were cell part (61 proteins), organelle (56 proteins) and organelle part (34 proteins) (Figure 4B). 

### 3.6. KEGG Pathway Analysis of the DEPs 

To further predict the involving signaling pathways involved in the identified DEPs, Kyoto Encyclopedia of Genes and Genomes (KEGG) pathway enrichment analysis was performed [20]. The results demonstrated that the 241 identified DEPs were classified into 7 pathways, including PI3K-Akt signaling pathway, focal adhesion, complement and coagulation cascades, tight junction, regulation of actin cytoskeleton, extracellular matrix (ECM)-receptor interaction, and carbohydrate digestion and absorption (Figure 5A). The DEPs were mainly enriched in the PI3K signaling pathway and complement and coagulation cascades (Figure 5B). 

### 3.7. Protein–Protein Interaction Network 

To elucidate the functional interactions among the DEPs, we performed protein–protein interaction (PPI) network analysis with the DEPs. As shown in Figure 5C, the DEPs were mapped to two major functional interaction networks, which were composed of two groups of strongly interacting proteins: POP1-SURF6-RPL7L1-ISG15-CCL5-C5-C9-CFB-F5-APOB-IGFBP7, which are associated with innate immunity; and INCENP-KIF20A-PCLAF-ISG15-CCL5-IL18, which are associated with the cell cycle and cellular components. Of note, at least four proteins were identified as hub proteins in these two tightly connected networks: ISG15, CCL5, GRWD1 and C5 (Figure 5C). For the two networks, there are five strong protein interactions in response to PRV infection, namely, ISG15, CCL5, C5, C9 and AFP, which are related to the immune response and PRV replication. Taken together, these data revealed that many of the proteins have different functions and can be connected by modulating the protein interactions, a finding that is expected to provide valuable insights for a better understanding of PRV pathogenesis.

## 4. Discussion

Zhao and colleagues applied iTRAQ labeling coupled with LC/MS to identify the cellular proteins expressed in PRV-infected PK15 cells [14]. In our previous study, we provided the first strong evidence that ISG15 is an efficient host effector against PRV infection in PK15 cells [17]. Numerous studies have ascribed an antiviral role to ISG15, including in the defense against porcine reproductive and respiratory syncytial virus [21,22], classical swine virus [23], and influenza virus [24]. It is reported that ISG15 deficiency could induce type I interferon production owing to defective negative regulation of IFN-I signaling as well as enhanced antiviral protection [25]. To date, the molecular role of ISG15 on PRV replication has not been reported. Considering that PK15 cells are an appropriate model for the study of host-PRV protein interactions [26,27], we chose to use the ISG15 knockout PK15 cell line (ISG15^−/−^-PK15) for the proteomic analysis to obtain experimental data that could identify the key cellular proteins involved in PRV pathogenesis. In our study, we performed a comprehensive analysis of the altered protein expression in ISG15^−/−^-PK15 cells responding to PRV infection by a quantitative proteomic approach of TMT-based labeling coupled with LC–MS/MS quantitative proteomics approach. A total of 241 DEPs were detected and quantified in PRV- and mock-infected ISG15^−/−^-PK15 cells at 24 hpi, of which 162 proteins were observably upregulated and 79 were downregulated. Through the verification of seven respective proteins by RT-qPCR and Western blot analysis, we found that the expression ratio between the mock and infected groups were in accordance with the TMT-based results. These DEPs were involved in numerous biological processes and signaling pathways, and these findings could provide insight to facilitate further analysis of PRV pathogenesis. 

PRV pathogenicity depends on the interactions between PRV proteins and host proteins, suggesting a crucial role of viral proteins and host responses in PRV pathogenesis. Similar to other alphaherpesviruses, PRV can establish a persistent infection partially because of its ability to escape the host immune response, especially the effects of type I interferon (IFN) [28]. ISGylation is a novel posttranslational modification, in which ISG15 is covalently conjugated to target proteins via the sequential action of three enzymes: the E1 activating enzyme UbE1L; the E2 conjugating enzyme UbcH8; and the major E3 ligase, Herc5 [29]. ISGylation is a reversible process that requires ubiquitin-specific protease 18 (USP18) to remove ISG15 from the target protein [30]. Notably, the proteomic analysis in this study showed that PRV infection markedly upregulated the expression of Herc5. Herc5, induced by the type I IFNs or viruses, functions as an ISG15 E3 ligase to promote overall ISGylation in response to viral infection and plays a role in host antiviral response [31]. We speculated that the upregulation of Herc5 might participate in the defense against PRV infection. Whether Herc5 plays an important role in the pathogenesis of PRV remains to be further identified. 

In this study, bioinformatics analysis indicated that the DEPs were mainly involved in metabolic processes, immune system processes, energy metabolism, cellular processes, biological regulation, multicellular organismal process, catalytic activity, and so on, which were activated after viral infection. KEGG pathway network analysis showed the DEPs were mainly enriched in the following pathways: PI3K-Akt signaling pathway, focal adhesion, complement, and coagulation cascades. In addition, we observed that the DEPs were mapped to two major functional interaction networks, both of which were involved in the immune system. These findings help to clarify the interactions between PRV and the host and provide mechanistic insights into PRV-induced host immune escape. 

Heat shock proteins (HSPs) are highly conserved functionally related proteins and are divided into different families including HSP40, HSP60 and HSP90, based on molecular weight [32]. It is commonly noted that viruses hijack chaperone pathways to facilitate their propagation in host cells, and heat shock proteins (HSPs) are associated with antiviral activity in some cases [33]. Previous studies have shown that host- and virus-encoded HSP40 can inhibit the pathogenesis in a wide range of viral infections [34]. In our study, the expression level of Hsp40 was observably decreased in PRV-infected ISG15^−/−^-PK15 cells. Thus, we speculated that PRV might promote viral replication via Hsp40 downregulation. However, the role of HSP40 in immune regulation and antiviral processes following PRV infection needs to be further studied.

We found that IL-18 was also significantly downregulated after PRV infection, suggesting that IL18 downregulation might mediate the defense against PRV invasion. Strap, serine-threonine kinase receptor-associated protein, contributes to its regulatory functions in several cellular processes, including signal transduction, cell cycle progression, and transcriptional regulation [35,36,37]. Recent studies have reported that Strap, a novel regulator of TLR3 signaling, positively regulates both the NF-κB and IRF3 signaling pathways by promoting the assembly of signaling molecules following TLR3 activation [38]. Downregulation of Strap is indicative of PRV infection in ISG15^−/−^-PK15 cells and further research is required to investigate its function during PRV infection. Mccc1, methylcrotonoyl-CoA carboxylase 1, is essential for cellular antiviral responses and potentiates virus-triggered induction of IFN and ISGs [39]. The upregulation of Mccc1 revealed that innate immune responses were activated in PRV-infected ISG15^−/−^-PK15 cells to combat the invading virus. The analysis of DEPs identified in this study was primarily descriptive, therefore, further functional studies are be needed to elucidate the role of the identified DEPs in PRV pathogenicity.

## 5. Conclusions

This study provides the first systematic analysis of the proteomic profiles in PRV-infected ISG15^−/−^-PK15 cells using a TMT-based proteomic approach. Functional analyses revealed that the identified DEPs were involved in various signaling pathways and biological processes. The findings will provide valuable insights not only for further investigation of PRV pathogenesis and defense mechanisms of host cells but also for the development of new antiviral strategies.

## Figures and Tables

**Figure 1 genes-12-01557-f001:**
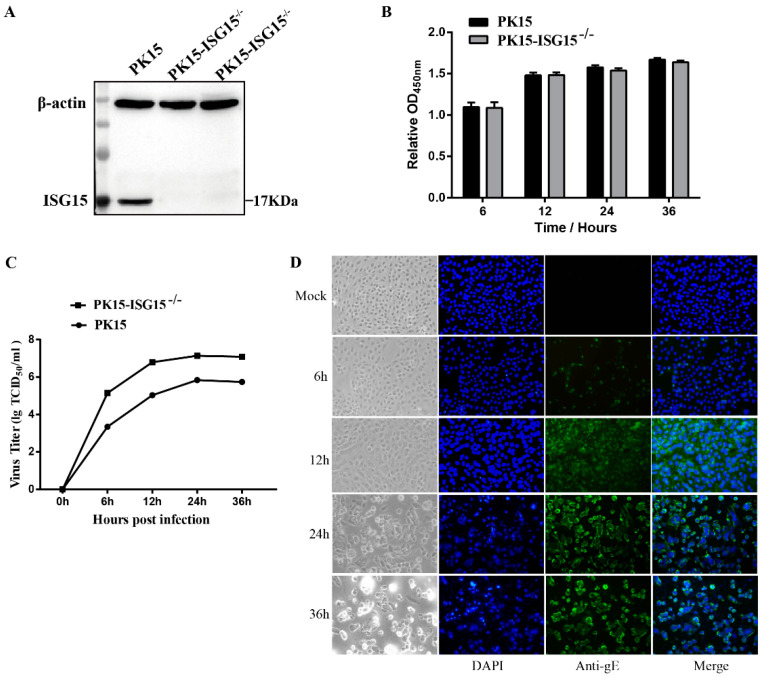
Establishment of the ISG15-knockout PK15 cell line. (**A**) Western blot analysis of ISG15 expression in PK15 cells and the ISG15 knockout sgRNA-1 and sgRNA-2 groups. (**B**) The effect of ISG15 knockout on PK15 proliferation was evaluated by an MTT assay. (**C**) One-step growth curve of PRV infection in ISG15^−/−^-PK15 cells at the indicated time points following viral infection. The viral titer is presented in TCID_50_/mL. (**D**) Confirmation of PRV proliferation in ISG15^−/−^-PK15 cells by IFA using the anti-PRV-gE mAb and Alexa Fluor 488-labeled goat anti-rabbit IgG (green); and mock-infected cells at 24 h was used as a negative control. Cell nuclei were stained with DAPI (blue) (400×). The data are presented as mean ± SD from triplicate experiments.

**Figure 2 genes-12-01557-f002:**
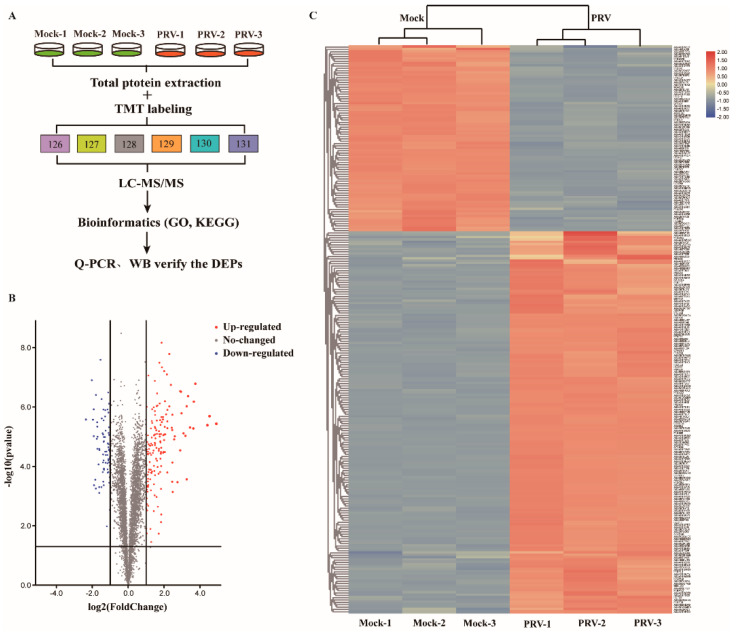
Induction of DEPs in ISG15^−/−^-PK15 cells following PRV infection at 24 hpi (*p*-value < 0.05, >1.2-fold change). (**A**) Strategy for quantitative proteomic analysis of PRV infection in ISG15^−/−^-PK15 cells. In all experiments were performed, the values are presented as the mean ± SD from three replicates. (**B**) Volcano plots of DEPs between mock- and PRV-infected groups. (**C**) Heatmap of DEPs based on hierarchical cluster analysis. Abundance profiles of proteins with distinctive features in the mock- and PRV-infected groups were calculated from the set of all 4958 proteins after normalization using hierarchical cluster analysis. Upregulated proteins are shown in red, and downregulated proteins are shown in blue.

**Figure 3 genes-12-01557-f003:**
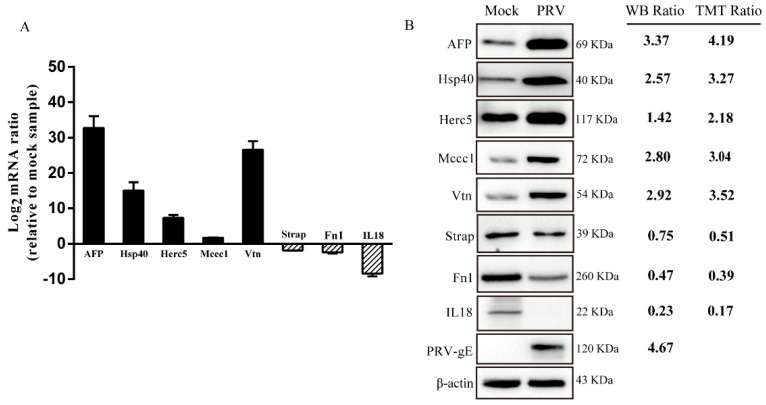
Validation of the randomly selected DEPs between mock- and PRV-infected ISG15^−/−^-PK15 cells. (**A**) Quantitative real-time PCR (RT-qPCR) analysis was performed to determine the relative mRNA expression levels of the randomly selected DEPs. (**B**) Western blot (WB) analysis of the expression of the selected proteins. The WB ratio and TMT ratio (infection/mock) are shown on the right side. The β-Actin was used as the housekeeping gene.

**Figure 4 genes-12-01557-f004:**
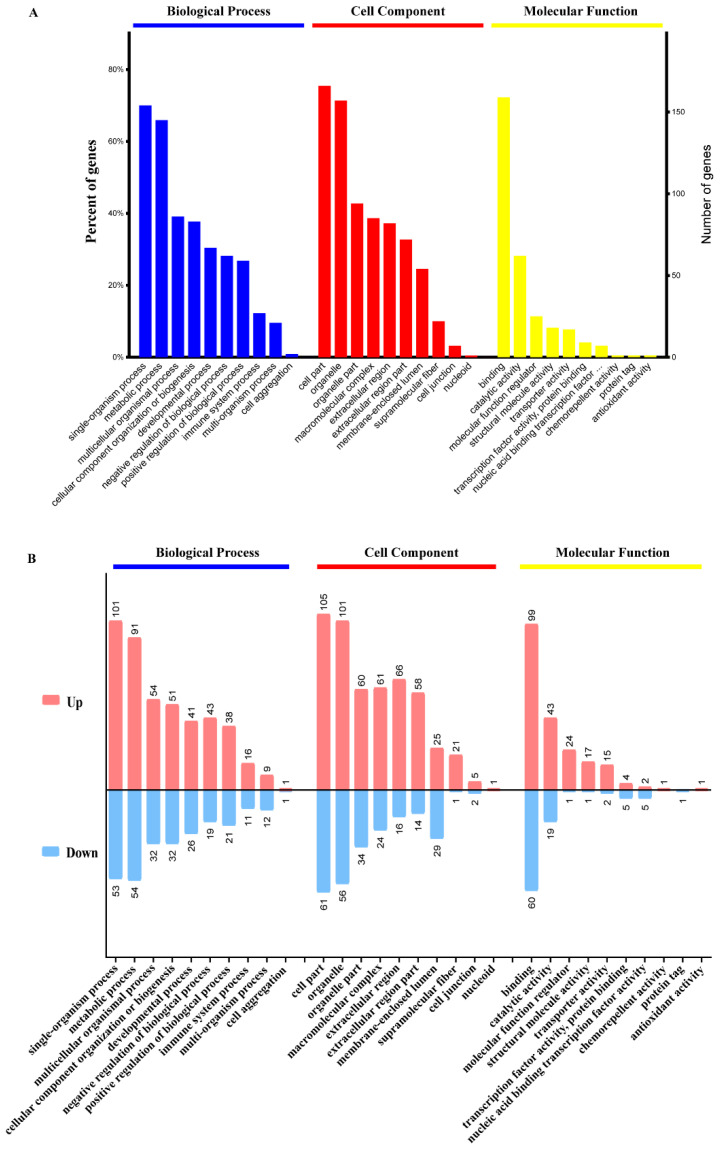
GO functional annotations of the DEPs. (**A**) Top GO terms in the biological process, cell component and molecular function categories enriched with the DEPs. The text on the abscissa indicates the name and category of the GO terms. (**B**) The pink and blue columns represent the upregulated and downregulated proteins respectively, with the number of altered proteins being marked at the top of each column.

**Figure 5 genes-12-01557-f005:**
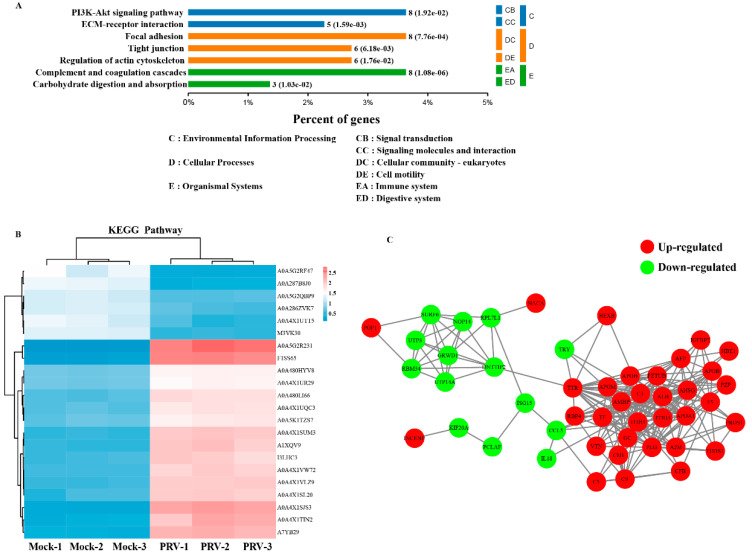
KEGG enrichment and PPI analyses of DEPs. (**A**) KEGG pathways significantly enriched (*p* < 0.05) in DEPs in the biological process category at 24 hpi. (**B**) KEGG pathway-based enrichment analysis. (**C**) A PPI network is presented to show the functional connections among the DEPs, and the red and green shadow represent upregulated and downregulated proteins, respectively.

**Table 1 genes-12-01557-t001:** Primers of the selected differentially expressed proteins.

Gene	Primer Sequence
Q-AFP-F	CAGGAAAATGGCGACCACAG
Q-AFP-R	GATGATAAGGTCAGCCGCTCC
Q-Hsp40-F	AACGACAGCGCAACAGTGA
Q-Hsp40-R	TGCTCCTCCTTTCAACCCTTC
Q-Herc5-F	ATGTGGGGACCAAACAGTGG
Q-Herc5-R	TGCGGTTATCGTGTTGGTGA
Q-Mccc1-F	GATGGGCTTGGAAGCAAAGAA
Q-Mccc1-R	TGCCTCATCTGCCATGTCTA
Q-Vtn-F	GTTTACCAGACGTGAGGCCG
Q-Vtn-R	CCTTGCACGACTCTTGGTCA
Q-Strap-F	TGCTACGCCAGGGAGATACA
Q- Strap-R	CAGCATCCCATACTTTGGCTGT
Q-IL18-F	AGCTGAAAACGATGAAGACCT
Q-IL18-R	CAAACACGGCTTGATGTCCC
Q-Fn1-F	AGAACCCTTGCAGTTCCGAG
Q-Fn1-R	GTCATCCGTGGGTTGGCTTA
Q-β-actin-F	TGGAACGGTGAAGGTGACAG
Q-β-actin-R	CTTTTGGGAAGGCAGGGACT
Q-PRV-gE-F	GACACGTTCGACCTGATGCC
Q-PRV-gE-R	TGGTAGATGCAGGGCTCGTA

## Data Availability

Not applicable.

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
