# Peer review of "Tandem Mass Tag-Based Quantitative Proteomic Analysis of ISG15 Knockout PK15 Cells in Pseudorabies Virus Infection"

_genes, 2021, doi:10.3390/genes12101557_

Round 1
Reviewer 1 Report
Comments to Author:
This report by He, Li, and co-workers establish a novel cell model to study ISG15 associated protein during PRV infection. They are not only establishing the cell model but also test associated proteins expression via various methods. These results provide solid data and justified conclusions. The findings in this report will substantially advance the field. I only have a few minor comments:
- Result 3.3, it will be better you could compare if there are any protein expression differences between PK15 WT cell line and ISG15-/--PK15 cell line. To confirm if ISG15 knockout will affect other proteins expression in PK15 cells.
- Did you confirm whether the knockout cells affect PRV infection or not? Like the infection ratio or virus attachment. Are they the same between PK15 WT cells and ISG15-/--PK15 cells?
- Figure 1D needs scale bar
- Figure 1D 6 h post-infection, the nucleus looks smaller compared to other time points. May need to select other figures.
- Line 221, it is better to write Table2 as Table S1, since the table is in supplemental data.
- Line 237, to verify DEPs, you must use quantitative reverse transcription PCR (RT-qPCR), not qPCR.
- Figure 5C, if you could, provide a high-resolution figure.
Reviewer 2 Report
The authors provide a study to uncover to significance of ISG15 in PRV infections and to provide insights into the relevant molecular mechanisms of ISG15 in PRV infection. The goal of the paper is interesting and could lead to novel insights in PRV biology and the role of ISG15. However I believe the paper has a major flaw in the choice of the control. Below I have outlined the major and minor issues with the current manuscript:
Major remarks:
The main issue with the manuscript is that a comparison is made between mock and PRV-infected ISG15-KO cells. By comparing samples with and without infection in the absence of ISG15, the role of ISG15 cannot be elucidated. To uncover the role of ISG15 in PRV infection, infected wild type PK15s should be compared with infected ISG15-KO cells. That way differences between the two conditions are due to the absence or presence of ISG15, i.e. the central question of the paper.
Many of the analyses and discussion thereof only mention DEPs. Since the biologically relevant entity in a cell are the proteins it would be better to discuss differentially expressed proteins instead of only the peptides. This is relevant for Figure 2B, 2C, but also for the analyses of Figure 4 and Figure 5.
In the discussion on line 364 it is mentioned that IL-18 is upregulated after PRV infection. According to Figure 3A and 3B, IL-18 is strongly downregulated upon PRV infection.
Minor remarks:
On line 12 in the abstract “the new targets against PRV” are referenced, but it is not clear to me what targets are meant
On line 83 in the materials and methods “the single exon” of ISG15 is mentioned. Porcine ISG15 has two exons. The exon which contains the coding sequence is probably intended, but this should be clarified
In section 3.1 the generation of the ISG15-KO cell line is described. It is mentioned that both clones were sequenced. If possible it would be relevant to describe what mutation was introduced. Was there a deletion or did a point mutation prevent the expression of ISG15 through a frameshift?
Also in section 3.1 on line 179. When it is mentioned that the cells were heterozygous for the ISG15 knockout it implies that the cells still contain one functional allele, however the western blotting controls show that this is not the case. What is meant with heterozygous here?
On line 201-202: “The time point when viral replication remains high is often regarded as the optimal time for proteomic analysis”. Please provide a reference.
In Figure 1C the one step growth curve of PRV infection in ISG15-KO cells is shown. It would be useful to show the growth curve of PRV in wild type cells in the same graph.
In the legend of Figure 2 it is mentioned that “downregulated proteins are shown in green” on line 232-233. There are not green dots or fields so this should probably be blue.
Round 2
Reviewer 2 Report
I don't think the experiment design used in this paper was the proper way to answer the research questions. However, since I shouldn't try to make the paper how I would write it, rather improve the authors paper, I have suggested accepting the manuscript.